



# ASSESSING SOIL FERTILIZATION EFFECTS USING TIME-LAPSE ELECTROMAGNETIC INDUCTION

Manuela S. Kaufmann[a], Anja Klotzsche[1], Jan van der Kruk[1], Anke Langen[1], Harry Vereecken[1], and Lutz Weihermüller[1]

[1]Agrosphere (IBG-3), Institute of Bio- and Geosciences, Forschungszentrum Jülich Germany

[a] former Agrosphere (IBG-3), Institute of Bio- and Geosciences, Forschungszentrum Jülich Germany

*Correspondence to*: Anja Klotzsche (a.klotzsche@fz-juelich.de)

**Abstract.** Adding mineral fertilizers and mineral nutrient is a common practice in conventional farming and fundamental to maintain optimal yield and crop quality, whereby nitrogen is the most applied fertilizer often used excessively, leading to adverse environmental impacts. To assist farmers in optimal fertilization and crop management, non-invasive geophysical methods can provide knowledge about the spatial and temporal distributions of nutrients in the soil. In recent years, electromagnetic induction (EMI) is widely used for field characterization, to delineate soil units and management zones or to estimate soil properties and states. Additionally, ground penetrating radar (GPR) and electrical resistivity tomography (ERT) have been used in local studies to measure changes of soil properties. Unfortunately, the measured geophysical signals are confounded by horizontal and vertical changes of soil states and parameters and the single contributions of those states and parameters are not easy to disentangle. Within fields, and also between fields, fertilization management might vary in space and time, and therefore, the differences in pore fluid conductivity caused directly by fertilization, or indirectly by different crop performance, makes the interpretation of large-scale geophysical survey over field borders complicated. To study the direct effect of mineral fertilization and its effects on the soil electrical conductivity, a field experiment was performed on 21 bare soil plots with seven different fertilization treatments. As fertilizers, calcium ammonium nitrate (CAN) and potassium chloride (KCl) were chosen and applied in three dosages. Soil water content, soil temperature, and bulk electrical conductivity were recorded permanently over 450 days. Additionally, 20 EMI, 7 GPR, and 9 ERT surveys were performed and at days of ERT measurements soil samples for nitrate and reference soil electrical conductivity measurements were taken. The results showed that the commonly used CAN application dosage did not impact the geophysical signals significantly. On the other hand, EMI and ERT were able to trace back the temporal changes in nitrate concentrations in the soil profile over more than one year. On the other hand, the results also showed, that both techniques were not able to trace the nitrate concentrations in the very shallow soil layer of 0 – 10 cm. Irrespectively of the low impact of fertilization on the geophysical signal, the results indicated that past fertilization practices cannot be neglected in EMI studies, especially if surveys are performed over large areas with different fertilization practices or crop grown with different fertilizer demands or uptake.





## 1 Introduction

To meet the challenges of a growing world population and to cope with the negative impacts of climate change, it is important to develop innovative and sustainable agricultural management strategies that increase crop yields, while maintaining healthy soils (Shah et al., 2019). Mineral nitrogen (N) or potassium fertilization is thereby essential in conventional farming to ensure high yields but also optimal crop quality. On the other hand, care must be taken to keep optimal timing and right amounts of

fertilizers to avoid pollution of ground- and surface-waters (Galloway et al., 2004; Bouwman et al., 2005; Olatuyi et al., 2012) and to reduce emissions to the atmosphere, which could negatively affect the climate (Butterbach-Bahl et al., 2013). The plant-specific nitrogen demand depends on the crop, while the required fertilization amounts depend also on the nitrogen replenishment from the soil (Drücker, 2016). Therefore, for sustainable agriculture with improved plant performance, it is highly important to develop tools for soil nitrogen mapping and monitoring, whereby those measurements should not be

restricted to the soil surface but should provide information on local heterogeneities within the root zone and to estimate the available fertilizer present in the soil profile. As stated by Kuang et al. (2012) precision agriculture explicitly considers field heterogeneities of soil properties and crop status by dividing the field into small-scale areas with same properties and/or crop development. This allows specific management of these sub-areas depending on their needs in terms of soil management (e.g., tillage or plant to be grown), fertilization, pest control, and irrigation (Hinck et al., 2016; Koch et al., 2023). Electromagnetic

induction (EMI) has been proven an excellent method to delineate crop management zones. For example, Hedley et al. (2004) performed EMI surveys of a pastoral–cropping farming system over a year and used these data to delineate zones of different apparent soil electrical conductivity (ECa). Further, the authors compared those zones with soil units of a conventional soil map and the results indicated that the ECa map related well to soil textural classes allowing grouping accurately different soil types. King et al. (2005) focused their study on the evaluation of yield maps and EMI data to determine management zones

within the fields. The results showed, that the management zones found by both methods provided useful information for the provisional delineation of soil type boundaries and crop management zones. A comparable study was performed by Serrano et al. (2022) who used EMI data and altimetry surveys in six experimental pasture fields to establish maps with three homogeneous management zones (HMZ) (less, intermediate, and high potential). The normalized difference vegetation index (NDVI), obtained from a proximal optical sensor and soil and biomass sampling were used to validate these HMZ. A step

further was presented by Brogi et al. (2019) who used EMI data and a supervised classification scheme to establish together with soil profile information and soil textural data a high-resolution soil map of a small agricultural area in Germany. In a follow-up paper, Brogi et al. (2021) used this map to simulate crop growth in the area and compared this simulation results with simulations based on available soil maps. For the validation of the simulation results leaf area index (LAI) data from remote sensing were used. The results showed that the EMI based soil map outperformed the use of the traditional soil maps

in the area studied. Recently, new avenues have been opened in the use of EMI, where either new interpretation approaches have been employed or EMI data have been combined with other non-invasive sensor data. For example, in a study by O'Leary et al. (2024) EMI data gathered were clustered with a neural network to demonstrate the correlation between soil electrical





conductivity with soil texture. The results showed, that the clustering outperformed the classical simple correlations between measured EMI and soil texture. As an example for sensor combination, van Hebel et al. (2021) combined ground-based EMI

and aerial crop data (NDVI) from drones to delineate field-specific management zones, which they interpreted with soil attribute measurements of soil texture, bulk density, and soil water content. Finally, they compared those zones with different yield and nitrate concentrations in the soil after potato (*Solanum tuberosum L.*) cultivation.

Even if all these studies successfully demonstrated the potential to derive soil properties or to delineate soil or crop management zones by geophysical methods such as EMI, EMI measurements also have limitations as the measurement of the apparent

electrical conductivity (ECa) is impacted by soil temperature (e.g., Corwin and Lesch, 2005), soil mineral surface polarization (chargeability) (Saey et al., 2013), profile soil water content (e.g., Huang et al., 2016), and pore water salinity (Triantafilis et al., 2000). Soil temperature effects on the EMI signal have been widely studied and correction formulas are proposed by e.g. Corwin and Lesch (2003) for EMI data measured over different times with variable temperatures. Additionally, differences in soil particle properties (e.g., clay content) and/or soil water content are often the targets being under investigation in EMI

surveys in non-saline soils (Kachanoski et al., 1988; Triantafilis and Lesch, 2005; Saey et al., 2009; Robinson et al., 2012; Rudolph et al., 2016) but confounding factors such as the impact of changes in pore water conductivity due to differences in fertilization or fertilizer uptake by different crop performance have been mostly neglected.

Commonly used organic or inorganic fertilizer releases charged molecules or ions (e.g., $NH_4^+$, $NO_3^-$, $Ca_2^+$, or dissolved organic carbon DOC) into the soil, affecting the soil electrical conductivity, and therefore, also the EMI signal. For example, Eigenberg

et al. (2002) observed that applied compost, manure, and mineral N fertilizer resulted in consistently higher electrical conductivities on arable fields and indicated the potential of EMI methods to provide reliable indicator of soluble N gains and losses. Eigenberg and Nienaber (2003) performed EMI measurements to identify areas of nutrient buildup beneath an abandoned compost site and the resulting ECa maps, or to be precisely the structures with high ECa, were in good agreement with the former row locations of the compost. Hereby, the identified historical compost rows showed significantly increased

soluble salts (1.6 times greater), $NO_3^-$ (6.0 times greater), and $Cl^-$ (2.0 times greater) compared with the area between the rows. Additionally, the authors also used yearly repeated EMI measurements to display annual changes associated to nutrient movement and transformations. In conclusion, Eigenberg and Nienaber (2003) stated that the correlation between EMI measurements and soil core analyses for $NO_3^-$, N, $Cl^-$, and EC provided ancillary support for the EMI methods. Kaufmann et al. (2019) showed, based on EMI measurement on a long-term field experimental sites with different fertilization and irrigation,

legacy effect on EMI data, offering new potentials in detecting and understanding the effects of agricultural management. Recently, Blanchy et al. (2020) used time-lapse EMI and electrical resistivity tomography (ERT) measurements to demonstrate their applicability to study water content changes induced by different commonly applied agricultural practices such as introduction of cover crops, compaction, irrigation, tillage and N fertilization. The results showed that different N application rates had a significant effect on the yield and leaf area index but only ephemeral effects on the dynamics of electrical





conductivity, mainly after the first fertilizer application. As the EMI signal was mainly impacted directly after N-fertilization one can raise the hypothesis that even common fertilization rates directly impact the measured ECa signal substantially.

Even if there are indications that commonly used fertilization rates impact EMI ECa measurements to an extend which is not negligible in the interpretation, comprehensive studies of the impact of varying fertilizer applications on EMI are currently lacking. Therefore, we intent in this study to close this gap by analyzing the effect of different N and potassium fertilization

rates on the measured EMI signal using time-lapse data measured over differently fertilized plots. For a better interpretation, the EMI measurements were accompanied by ERT and ground penetrating radar (GPR) measurements as those systems can provide additional information on water content (GPR) or highly resolved vertical EC information (ERT). For ground truthing, soil samples were extracted, and the nitrate content and soil bulk electrical conductivity ($EC_e^{Soil}$) were measured at times of geophysical surveys. Finally, sensors automatically recorded the bulk soil EC, soil water content, and soil temperature at

different depth to help interpreting the measured geophysical data.

## 2 Materials and Methods

### 2.1 Selhausen test site

The experiment was conducted at the test site Selhausen, Germany (Figure 1) which is part of the TERENO (TERrestrial ENvironmental Observatories) Eifel-Lower Rhine observatory in North Rhine-Westphalia, Germany (Pütz et al., 2016; Bogena

et al., 2018). The site consists of quaternary sediments covered by loess and is located at a transition zone between the Upper and Lower Terrace of the Rhine/Meuse river system. The main textural fraction is silt with 55 - 67% silt in all horizons (Weihermüller et al., 2007). Annual precipitation is 715 mm and mean annual temperature is 10.2 °C (Rudolph et al., 2015). The ground water depth shows seasonal fluctuation, but no groundwater was detected in the first 3 m of the subsurface. In the last years, several geophysical studies have been performed at this test site (e.g., Jonard et al., 2011; Busch et al., 2014; von

Hebel et al., 2014; Kaufmann et al., 2020) and based on these previous geophysical studies, a location at the lower terrace was chosen as best suited for the experiment as low natural soil heterogeneity was expected there (blue box in Figure 1a). In order to prove low soil heterogeneity and uniform soil electrical conductivity (ECa), an EMI survey was performed on 2nd of March 2017 (results shown in Figure 1b and c). Based on those maps the experimental plots were established in the southern end of the field site characterized by low variability in soil apparent conductivity (black box in Figure 1b and 1c).

### 2.2 Field experiment

To measure the effect of different fertilizer and dosage over time on the EMI measured ECa data a field experiment with seven different treatments were established. The treatments consisting of a control (no fertilizer applied), two different commercial fertilizers, calcium ammonium nitrate (CAN) with 26% N-content and potassium chloride (KCl) with 40% K. For the CAN and KCl treatment, three different dosages (normal (a), double (b), and ten-fold (c)) were selected. A normal N fertilization



level was defined as 190 kg N/ha, resulting in application dosage of 0.67, 1.33, and 6.67 kg per plot for (a), (b), and (c), respectively. KCl was dosed equally for (a) and (b), but for treatment (c) the amount had to be reduced to 5.33 kg per plot as this was the maximum amount to be diluted in the water used for irrigation. Each of the seven treatment was triplicated and randomly assigned to the 21 plots of 9 m² (3x3 m) in size. Plots were separated 1 m from each other (see Figure 2). The fertilizers were dissolved in water (10 to 30 liters depending on fertilizer dosage) and homogeneously applied with commercial

watering cans on the 3rd April 2017. This date is later used as reference day 0 and all following days are denoted as day after fertilization (DAF). Over a time period of 485 days (DAF 485) EMI measurements were performed to monitor the effect of fertilization over time (see Table 1). As mentioned, also GPR and ERT measurements were performed to support the interpretation of the fertilizer effects on the EMI measured ECa.

For permanent record of the soil water content (m³/m³), soil temperature (°C), and bulk electrical conductivity (mS/m) 20

sensors (5TE, 5TM, and ECH2O-TE) from Decagon Devices Inc. (Pullman, WA, USA) were installed. The installation depths of the sensors were 10, 20, 30, 40, and 60 cm, whereby always four sensors were installed at each depth with a separation of 10 cm. As not all sensors measure EC, care was taken to install at least two sensors capable of measuring EC in each depth. The sensor data were logged hourly and daily averaged afterwards. In order to guarantee error-free geophysical measurements and avoid N uptake by plants, the plots had to be cleared of vegetation. Therefore, the area was treated three times with

glyphosate-containing pesticide. One application took place before the trial, one in June 2017, and the last one in June 2018.

### 2.3 Climate data

Air temperature and precipitation are directly affecting soil temperature and soil water content (SWC), and therefore, indirectly the geophysical signals measured. Climatic conditions were recorded by the nearby TERENO climate station SE_BK_002 (http://teodoor.icg.kfa-juelich.de), located about 50 m south from the experimental plots (see Figure 1a). For a better

illustration, the air temperature is plotted as daily mean and the precipitation is provided as the daily sum in mm (Figure 3). Overall, the temperature followed typical seasonal pattern with air temperatures ranging from -5°C to 30° C. In the first 90 days, dry weather conditions with low precipitation were observed. To accelerate the downward leaching of the fertilizers an additional irrigation of 39 mm was applied 10th July 2017 (DAF 98, highlighted in black). After the irrigation, precipitation increased and remained normal until summer 2018 (DAF430).

### 2.4 Geophysical data acquisition and data processing

### 2.4.1 Electromagnetic induction (EMI)

EMI measures the apparent electrical conductivity (ECa) as a weighted average value over the vertical bulk electrical conductivity $\sigma$ distribution within a certain depth volume (Keller and Frischknecht, 1966). The weight average per depth (Figure 4a), depends mainly on the coil separation $s$ and orientation (McNeill, 1996). Coils can be either oriented vertical co-

planar (VCP), which results in more sensitivity to shallow soil depths, or horizontal co-planar (HCP) resulting in increased





sensitivity at greater depths (Figure 4b). The so-called depth of investigation (DOI) is defined as the depth interval in which up to 70% of the relative response function accumulates. For the VCP and the HCP this is at around 0.75 and 1.5 times $s$, respectively (McNeill, 1980). By investigating the relative sensitivity curves for VCP and HCP orientation normalized to the coil separation $s$, it is clear that the VCP mode is most sensitive to the shallow surface and becomes less sensitive with increasing depth. In contrast, the HCP mode is less sensitive to shallow surface and peaks at a depth of around 0.4 times the coil separation $s$ (Figure 4a).

The EMI measurements were carried out with a rigid-boom multi-configuration CMD-MiniExplorer (GF-Instruments, Brno, Czech Republic.) mounted on a crutch. This system measures with 30 kHz and has three receiver coils ($s$ = 0.32, 0.71, and 1.18 m) and was used in VCP and HCP configuration. Resulting approximately in depth of investigation ranging from 0-25 cm up to 0-118 cm. On all measurement days, the 21 plots were measured four times at different location within the plot to exclude potential small heterogeneities within a plot. Therefore, the instrument was rotated by 45° horizontally around the plots center (see Figure 5b). Each measurement was averaged over 100 readings while keeping the instrument stationary and directly on the surface (Figure 5b). All EMI data were processed with custom made MATLAB codes. As raw EMI measurements are often prone to systematic errors (e.g., Gebbers et al., 2009; Nüsch et al., 2010), the raw ECa$^{EMI}$ were only used for qualitative analyses over time for the same configuration.

### 2.4.2 Electrical resistivity tomography (ERT)

ERT measures the soil electrical resistivity using galvanic coupling. While at two electrodes current is injected to the ground, the electrical potential difference is measured between two other electrodes. The measured apparent resistivity is an average resistivity over a certain depth and space, depending on the electrode spacing used for the measurement. By measuring different combination along a transect and followed data inversion, a 2D profile of the soil resistivity can be achieved (Binley, 2015). In this study, the ERT data were converted to its reciprocal soil electrical conductivity for final analysis and plotting.

The ERT measurements were carried out on nine dates along the same two transects as used for the GPR measurements (see Figure 2). Therefore, two 30 m long transect were acquired using an electrode spacing of 25 cm and Dipole-Dipole setup. Measurements were taken by a Syscal Pro ERT system (IRIS Instruments, Orlean, France) (see Figure 5a). Measured data were inverted using the open access software BERT (https://www.pygimli.org/, Rücker et. al, 2017) with a predefined mesh.

### 2.4.3 Ground penetrating radar (GPR)

GPR emits electromagnetic (EM) waves, which are reflected or refracted in the soil at changes of either the soil dielectric permittivity ($\varepsilon$) or electrical conductivity ($\sigma$). The dielectric permittivity and electrical conductivity can be related to the propagation and the attenuation of the EM wave, respectively. As the fertilizer increase the electrical conductivity of the subsurface it is expected to see differences in the GPR measured signal attenuation and amplitudes. One common GPR measurement setup is the common-offset profiling (COP), where transmitter and receiver antennae are moved along defined





profiles with a constant spacing between the antennae. This method allows to identify structures in the subsurface. To convert the data measured in time to depth, the measured profiles can be time-to depth corrected with literature values or point measurements (Jol, 2009).

At seven different days (see Figure 3) two transects spanning over plot 1 to 7 and plot 15 to 21 (see Figure 1) were measured with 500 MHz PulseEkko GPR antennas from Sensors & Software Inc. (Mississauga, Canada). The 30 m long profiles were measured with a common offset of 0.23 m between the transmitter and receiver. The data are standard processed (dewowed, time zero correction, and gain), cut at 1 m depth and as additional step Hilbert enveloped transformed to better visualize amplitude, and hence, electrical conductivity changes. More details about the GPR post processing can be found in Dal Bo et 195 al. (2019).

### 2.4.4 Temperature correction of EMI and ERT data

To account for temperature effects on the electrical conductivity and to compare the measurements over time, inverted ERT ($EC^{ERT}$) and EMI ($ECa^{EMI}$) data were standardized to a reference soil temperature of 25°C using the approach of Corwin and Lesch (2005):

$$EC_{25} = EC_T \cdot \left(0.4470 + 1.4034\, e^{(-T/26.815)}\right) \tag{1}$$

where $EC_T$ is the electrical conductivity measured at a particular soil temperature $T$ in °C. In our case the soil temperature of the in-situ sensors was used, while the temperature of 60 cm depth was assumed to be valid for deeper depths too. For the $ECa^{EMI}$ values, the soil temperature was weighted average based on the depth sensitivity of the configuration (Blanchy et al., 2020).

### 2.5 Soil sampling and chemical analysis

For validation purposes, soil samples were collected over the time of the experiment at the same dates as the ERT and GPR measurement were taken, resulting in nine ground truth sampling days (see Table 1). The samples were taken by Pürckhauer augering to a maximum depth of 100 cm. The soil cores were divided into predefined depth intervals of 0-10, 10-20, 20-30, 30-40, 40-60, 60-80, and 80-100 cm. The soil was stored in plastic bags at a temperature of 4°C and then dried for 48 h at 105°C and homogenized. The soil bulk electrical conductivity ($EC_e^{Soil}$) was measured with a mixture of 1 part soil and 5 parts 210 distilled water. Note, that the obtained $EC_e^{Soil}$ values are already temperature corrected. Nitrate content ($NO_3^-$ [mg/kg]) were measured photometrical according to DIN 38405-9 (Deutsches Institut für Normung, 2011) on a mixture of 1 part soil to 3-part distilled water. The photometrically measured liquid concentration was back-calculated to the weight of the extracted soil and normalized to mg nitrate $NO_3^-$ per kg soil. The nitrate concentration was only measured on plots where nitrate fertilizer has been added and due to expected slow vertical translocation of the fertilizer after application only the shallow samples have 215 been measured at the first sampling dates.





## 2.6 Statistical analysis

All obtained nitrate contents, $EC_e^{Soil}$, $ECa^{EMI}$, and $EC^{ERT}$ data, were assigned to the corresponding plot number, and DAF. For the ERT data, the inverted conductivity was extracted from the profiles for each plot and depth interval (same depth interval as soil samples) ($EC^{ERT}$). For this, the ERT derived EC data were extracted from the profile data over the defined sampling
intervals. For the EMI data, measurements in both VCP and HCP mode were considered. As a first analyzing step, it was checked if the seven treatments differ significantly to the control for each individual measurement day using a Mann–Whitney $U$-test, with a Kruskal–Wallis $H$-test to identify significant differences between means at a probability of $p<0.05$). Additionally, the mean ($\mu$) and standard deviation ($\sigma$) per plot was calculated, and out of it the coefficient of variation (CV) $CV = \sigma / \mu$ was calculated to check the variation within the treatments. The Pearson correlation coefficient $r$ was used to describe the
correlation between soil characteristics and geophysical values. The corresponding tables to the statistical analysis can be found in the supporting information.

# 3 Results and Discussion

## 3.1 Soil sensor data

The soil temperature follows the typical seasonal pattern over all depths (see Figure 6a). At the start of the experiment, the
SWC (see Figure 6b) was in the range of 0.26 to 0.29 m³/m³, but as a consequence of low rainfall and warm weather conditions the soil dried out subsequently and the SWCs decreased to 0.21 m³/m³ at 10 cm depth at DAF 60-85. After irrigation on the10th of July 2017 (DAF 98) the SWC at all depths rapidly increased and remained relative stable until DAF 420. As can be seen, most dynamics was in the shallow sensors (10-30 cm) measuring small fluctuation caused by rain events and dry downs but also a large drop around DAF 330 is detectable, which is associated to snow coverage in the field. Because of the heat wave
in summer 2018, the SWC decreased from DAF420 until the end of the experiment. The soil dried out the most in the shallow depth with a minimum SWC of 0.16 m³/m³.

The soil bulk electrical conductivity ($ECe^{ref}$) [mS/m] shows a similar behavior as the SWC sensor data, including the decrease during the two drought periods. The deeper two sensors (40 and 50 cm) remain almost stable between 20-25 mS/m throughout the entire year. Here it has to be noted, that the sensors were installed outside the plot experiments, and therefore, only reflect
the EC situation of the control plots.

## 3.2 Geophysical measurements

### 3.2.1 Changes across treatments

In a first step, we compare exemplarily the measured GPR and ERT data across the different treatments from transect 2 (see Figure 2). Before the application of the fertilizers (DAF 0) a relatively homogeneous subsurface with 2 layers can be identified



for both the ERT and GPR data as shown in Figure 7a and g, which agrees well with the EMI maps, which were performed prior the plot setup (Figure 1b and c).

In the ERT transect, a low conductivity layer of 5-10 mS/m up to a maximum depth of 50 cm overlying an intermediate EC layer with small internal variation can be observed. The shallow low electrical conductivity layer can be associated with the plough horizon, which normally has a lower porosity as the underlying layer (Jeřábek et al., 2017). After fertilization, a clear
change in the $EC^{ERT}$ values across the transect are detectable. As expected, the largest changes can be observed at the plots with highest (the ten-fold of the normal N dosage of N and KCl - treatment Nc and KClc) dosage applied. The ERT images after 35, 66, and 135 days after application (DAF) show an increase of $EC^{ERT}$ up to 100 mS/m up to a depth of approximately 40 cm. Over time, the higher $EC^{ERT}$ values move downward and reached at DAF 485 a depth of 100 cm with an $EC^{ERT}$ of about 60 mS/m. In contrast, the low dosage (normal recommended N dosage) of N and KCl (KCLa and Na) show only minor
changes over time in the ERT derived EC, while the doubled dosage KClb and Nb indicate changes in the $EC^{ERT}$ until DAF 135 in the shallow depths, while at later times the values are smeared over the depth and no clear effect is optically visible. The reference plot remains relatively stable over time and shows only minor changes, which are mainly related to changes in soil water content as a result of precipitation and evaporation.

In the GPR profile gathered before fertilizer application, also a homogeneous subsurface can be seen, although, small variation
of the envelope amplitude in the first 5 ns can be detected. Here, it has to be noted, that the GPR signal is attenuated fast because of the high clay content of the soil, and therefore, deeper subsoil information cannot be obtained, and the information is mainly restricted to the first 20 cm of the ground. Nevertheless, changes in the radargram over time can be detected. For example, at DAF 66 a very strong effect of the high fertilization dosage can be seen at the location KClc and Nc not only in the GPR radargram but also in the calculated Hilbert envelop images. In contrast, at DAF 400, the GPR transect is almost
homogenous as it has been before the fertilizer application, indicating that the fertilizers have been leached to deeper zones and cannot be captured by the GPR or that the concentrations have been diluted to such an extent that they were not detectable anymore. Although, amplitude changes were detectable for the high fertilizer application dosages, GPR data were not further interpreted as it was found difficult to disentangle various effects on the GPR data (e.g., water content changes) without additionally information about those changes and more advanced data processing such as full-waveform inversion (Liu et al.,
2018) or multi-offset GPR (Kaufmann et al., 2020).





### 3.2.2 Spatial and temporal changes at the point-scale

For a more detailed analysis the temperature corrected EMI and ERT as well as the soil samples were sorted by depth and treatment over time and all treatment replicates were averaged (arithmetic mean) and discussed in the following. In Appendix Table 1, significant ($p<0.05$) differences of the treatments from the control are also listed.

*EMI measurements*

Prior fertilization on the 3$^{rd}$ of April 2017 (DAF 0) ECa$^{EMI}$ for the shallowest depth of investigation (VCP 0.32 m DOI = 0-0.25 m) was the lowest with 10.4 mS/m and highest for the deepest configuration (HCP 1.18 m DOI=1.78 m) with 21.8 mS/m, indicating a higher SWC in the deeper soil profile (Figure 8). After fertilization, a clear trend in the measured ECa$^{EMI}$ can be seen except for the deepest sensing configuration (HCP 1.18 m) with increase ECa$^{EMI}$ compared to the control. Especially, the

increase of the higher dosages (b and c treatments) for the two shallow measurement configurations VCP 0.32 m and HCP 0.32 m (DOI = 0.45 m) is quite large with up to 40 mS/m. For the deeper configuration and later times, only the high dosages of N and KCl showed elevated ECa$^{EMI}$. After fertilization, ECa$^{EMI}$ increases to highest values at DAF 49, with a noticeable drop afterwards, explainable by the low SWC and corresponding lower soil bulk electrical conductivity (EC$_E^{SOIL}$) as shown in Figure 6c. After the irrigation and precipitation, the ECa$^{EMI}$ increased again and remained high until DAF 195. A smaller drop

in ECa$^{EMI}$ can be found at DAF 310 for VCP 0.32, HCP 0.32 and to a less larger extend also in VCP 0.71, whereby this drop is earlier in time as the sharp drop in SWC and EC$_E^{SOIL}$ found in the senor data depict in Figure 6c. Having a closer look into the sensor data, one will see a small drop in SWC measured at 20 cm depth, which seems to be associated to a short period of dryer soils and an associated drop in ECa$^{EMI}$. Over the complete time span the coefficient of variance was except few outliers for all treatments and configuration below 0.4, whereby for the smallest coil configuration (VCP and HCP 0.32) in general

more spread (between 0.2 and 0.4) is present than in the other configurations (mainly below 0.2). Additionally, for ECa$^{EMI}$ significance in the differences between the treatments and the control was calculated and it can be seen that the ECa$^{EMI}$ for the normal dosage Na never differed significantly from the control. Whereas for KCla low significant difference ($p>0.2$) were present between DAF 20 and 135 except for the days with low SWC (DAF 66-91) for the VCP configuration. Also in the double dosage (b) significantly higher differences can be found within the KClb treatment compared to Nb. Hereby, Nb is

mainly significantly different to the control at $p<0.05$, whereas for KClb it is mainly different at $p<0.01$ after DAF 35 and until DAF 193 for all three VCP configuration and HCP 0.32 and HCP 0.71. Only the dates between DAF 66 to 91 for the double treatment Nb are not significantly different from the control, whereby those dates are associated to low SWC in the shallow soil. In contrast, plots treated with KClb are less affected by the SWC drop and still significantly different to the control. For the deepest sensing configuration HCP 1.18 no significant difference was measured at any time of the experiment. The highest

dosage (Nc and KClc) differed significantly ($p<0.01$) for all 3 VCP configuration and HCP 0.32. For HCP 0.71 high to medium significant ($p<0.5$) difference to the control was found except for DAF 66 for KClc, whereas the deepest sensing configuration HCP 1.18 only started to differ significantly after DAF 102 (except for DAF 135) for Nc and after DAF 310 for KClc. This





pattern indicating that the fertilizer slowly moved downwards over time. Here, it has to be noted, that for this analysis only the non-calibrated EMI data were used as those data are classically available and easiest and fastest to acquired. On the other hand,

those EMI data do not allow for inversion of depth specific bulk EC, which might improve the overall analysis.

*ERT measurements*

In comparison to the non-calibrated and non-inverted EMI data, depth specific data can be used for the ERT as shown in Figure 7 for the intervals 0-10, 10-20, 20-30, 30-60, 60-80, and 80-100 cm. Overall, the ERT derived electrical conductivity (EC$^{ERT}$) followed a similar trend as the EMI data especially for both high dosages (Nc and KCLc) applied, whereby one can see that

the EC$^{ERT}$ is mostly affected by the fertilization application in the top layers over all sampling dates. One can also clearly detect the downward movement of the fertilizers over time, which is easily to follow for the high CAN dosage (Nc), where the EC$^{ERT}$ was highest for early measurement days in the top layer up to a depth of 30 cm, whereas for the later measurement days (DAF 310 and after) the EC$^{ERT}$ is higher for the layer 10 - 20 cm compared to those measured for the first layer (0-10 cm). Here, it has to be noted, that the control plot did show smaller changes in measured EC$^{ERT}$ over time, whereby these changes

are only caused by changes in SWC. Even tough, the daily covariance was relatively low (>0.4, except few outlies) over the course of the experiment, for the highest dosage (Nc and KClc) largest spread in EC data over depth can be observed. Also for EC$^{ERT}$ significance in the differences between the treatments and the control was observed. For the normal dosage Na only few significant ($p<0.05$) difference are present directly after the fertilization application at DAF 8 at a depth between 20-40 cm and between 80- 100 cm. For the deeper depth also significant difference were found for DAF 102-193. For Nb, KCla, and

KClb similar pattern were observed with high significant differences at shallow depth (0-10 cm) only for DAF 36, for 10-40 cm from DAF 36 to 193 except on DAF 66 for depth 40 cm and for 60-100 cm between DAF 102 and 193. After DAF 193, no significant difference to the control was found for the normal and double dosage, (except at 10 cm at DAF 193 for Nb). For the highest dosage Nc and KClc, for depths between 30-100 m high significant difference ($p<0.01$) with respect to the control was observed from DAF 36 until end of experiment (DAF 485), whereby for the shallow depth (0-20 cm) no significant

difference was found at DAF 310 and 400. However, for DAF 410 and 465 significant differences are present again for the application with the ten-fold dosage. Similar to the EMI results the ERT shows downward leaching of the fertilization over time, but better depth resolved as the data obtained by EMI. This result could be expected, as ERT is capable to vertically resolve EC differences with high spatial resolution and at low EC differences as stated by Garré et al. (2011). On the other hand, for the normal and double dosage the fertilizer leached deeper than 20 cm within the first two months leading to

dispersion of the masses and concentrations (Ellsworth and Jury, 1991), and therefore, also to lower impact on the gathered geophysical signals. As a consequence of further translation, spreading of the fertilizer plume and the difference to the control become statistically irrelevant (similar to those shown for the EMI results). However, this also shows that normal fertilization rates clearly impact geophysical measurements at least a certain time after application, and therefore, should not be neglected if geophysical measurements are performed over different fields not managed the same way or if different areas within the

field are fertilized differently or if plants take up different amounts of fertilizers due to variable plant growth induced by e.g.,





non-homogeneous water supply. Finally, it has to be noted, that both EMI and ERT measurements showed less difference between the treatments and the control at days with low SCW (for example DAF 66 or 193), which is in line with previous findings by Schmäck et al. (2022). In consequence, this means that optimal SWC conditions are required for EMI and ERT surveys to identify differences in fertilization states of the soils, whereby most suitable are conditions with relatively high SWCs.


*Soil nitrate concentrations and soil bulk electrical conductivity*

For further interpretation of the geophysical derived information the nitrate concentration and the soil bulk electrical conductivity ($ECe^{Soil}$) data analyzed in the laboratory on the soil samples were used (Figure 9 b and c). As the nitrate analysis is time consuming only those depths were analyzed where elevated nitrate concentrations have been measured in the previous

measurement days plus the next underlying sampling layer. If this next layer also showed higher concentrations as the reference (reference = concentration prior fertilizer application) the next depth was also analyzed. This simplification can be made if preferential water flow and associated preferential solute transport can be excluded and only slow matric flow of the solutes is assumed. For those layers not analyzed for the corresponding sampling day, the nitrate concentration measured prior fertilizer application (DAF 0) was used. Additionally, KCl was also not measured as the extraction and measurements are also extremely

tedious and costly and overall focus of the study was on N-fertilization effects.

Overall, the $ECe^{Soil}$ and nitrate concentrations follow a similar trend as the EMI or ERT data especially for both high dosages of N and KCL. The mean $ECe^{Soil}$ in the control plot varies to a small extend between 3 and 15 mS/m in the first 20 cm over time, probably affected by weather events and translocation of the background solute and/or mineralization of organic matter. At greater depth, the $ECe^{Soil}$ decreases to values round 8 mS/m likely also due to lower organic matter content, and therefore,

lower release of DOC. The mean soil nitrate concentrations in the control plot varies also to a small extend between 5 and 15 mg $NO_3^-$ per kg soil with some variability over the nine measurements dates (see Figure 9) likely caused by the small scale heterogeneity of the soil and the small scale sampling via the Pürckhauer auger (see detailed discussion below). In general, the nitrate concentrations in the upper 60 cm of the control plots are always slightly higher as those in the deeper soil, which can be attributed to lower organic matter content at larger depth forming nitrate by decomposition. As already discussed for the

$ECe^{Soil}$ the temporal changes in nitrate concentration of the control plot where no N fertilizer was applied, can be sorely associated to the mineralization of organic matter stimulated by higher soil temperatures in the uppermost centimeters of the soil profile and nitrate release to the soil.

As expected, highest $ECe^{Soil}$ and nitrate concentrations can be found in the high dosage plots Nc and KLc. In general, the $ECe^{Soil}$ followed more or less the same pattern as already discussed for $EC^{ERT}$. Looking at the nitrate concentrations, one can

see that the nitrate concentrations increased rapidly after the fertilizer application up to ca. 1400 mg/kg in the first 10 cm until DAF 66 for the high dosage. After irrigation and further precipitation, the nitrate concentrations decreased in the upper 10 cm but remained still higher as those found in the control plot. Despite the high values found, no significant differences ($p<0.2$)



was observed for any depth and treatment for the normal and double dosage for ECe$^{Soil}$ and nitrate content with respect to the control plots. For the high dosage, significant difference ($p<0.1$) were partly present, for ECe$^{Soil}$ at 0-10 cm at DAF 102 and 195, and for KClc only at DAF 36 and 193 at depth 10-40 cm and for Nc and KClc from DAF 102 to 400 (except DAF 310 at 10-20 cm for Nc). For 80-100 cm depth the Nc treatment differed significantly from the control from DAF 310 - 465 and for KClc at DAF 102 and from DAF 310 – to 465, respectively. For the nitrate content for the N fertilized plots similar pattern as for ECe$^{Soil}$ can be observed with highest concentrations for the highest dosage especially for the shallow depth over the entire experimental period and a slow downward movement over time. Interestingly are also the higher concentrations found in the highest dosage plots (Nc) at later times, indicating a downward movement of the nitrate.

In general, the soil sample data (ECe$^{Soil}$ and nitrate concentrations) had the highest coefficient of variation (CV) of all measurements, which explains the weaker significant differences between treatments and control compared to ERT and EMI, which both showed lower CV. The high variability of the nitrate concentrations and ECe$^{Soil}$ of the soil samples can be partly explained by the sampling itself, as only few soil samples were taken (1 per plot, so 3 per treatments) compared to EMI (4 per plot, so 12 per treatment) and ERT (~5 per plot, 10 treatment) as well as the small volume sampled by the Pürckhauer auger. For the EC$^{ERT}$ at DAF 310 and beyond we speculated that the decrease of EC$^{ERT}$ is caused by downward movement of the nitrate to deeper soil layers. Looking at the ECe$^{Soil}$ data and nitrate concentrations we can see the same pattern of lower ECe$^{Soil}$ values in the upper soil layer (0-10 cm) and higher ones in the underlying layer. Same holds for the nitrate concentrations, As ECe$^{Soil}$ and the soil nitrate concentrations are independent on SWC changes, it only shows the actual status of ions available at a certain time in the soil, and therefore, proved the hypothesis of downward translocation.

### 3.2.3 Correlation between measured soil states

As it is known that geophysical measurements such as those gathered by EMI and ERT are influenced by various soil states and parameters such as soil texture (e.g., clay content), soil pore water salinity, and soil temperature (McNeill, 1980; Corwin and Lesch, 2005) one aim of this study was to disentangle the impact of fertilization on the measurement signal. Therefore, correlations between the different measured states were performed (Table 2 and S1-S4). As the soil texture will not affect the results measured in the different treatments, as soil texture is known to be very stable over time (Upadhyay and Raghubanshi, 2020), soil texture can be assumed as a static component in the analysis, and therefore, neglected.

In a first step, the sensor based SWC and ECe$^{ref}$ were correlated (Figure 10). Here, it has to be noted, that the correlation was done either for the entire dataset measured over the entire period or for the single days where geophysical measurements were performed. Before looking into details of the individual correlations, attention should be drawn to the effect of irrigation on the ECe$^{ref}$. Directly after the irrigation (blue dots in Figure 10) at DAF 98 the ECe$^{ref}$ showed higher values for respective soil water contents and dropped down to lower ones for the same water contents (yellow to red dots). This effect can be explained by the higher electrical conductivity of the irrigation water used (49.1 mS/m) compared to the rainwater (classically between 3 - 6 mS/m in the region) feeding the soil water under non-irrigated conditions. As shown by Kaufmann et al. (2019) the



irrigation does not only affect the soil EC over short times but can be traced back by EC measurements over longer periods. However, at a deeper depth (30 - 60 cm) the $ECe^{ref}$ is less affected by the irrigation as the irrigation water had to travels downwards and also had to mix with already existing pore water, and thereby, equilibrated in ion concentration.

Coming back to the correlation, it becomes visible that for the shallow subsurface between 0 - 30 cm depth a moderate correlation for both, the entire dataset and measurement days ($r > 0.6$) was calculated, whereby the effect of the irrigation
becomes less and less pronounced in the deeper soil layers as the points (bluish and red/yellow) get closer to each other. For depth at 40 cm the correlation over all days is slightly lower ($r = 0.43$) and at 60 cm only a low correlation ($r = 0.25$) for the entire dataset was found. The low correlation especially at 60 cm depth can be explained by the low variability of $ECe^{ref}$ and SWC. Considering only the days where geophysical measurements were performed, the correlation generally increased for all depth and exceeds $r$ values of 0.8 for the depth 10 to 30 cm. Even at lower depths $r$ stayed relatively high with $r = 0.7$ and $0.51$
for 40 and 60 cm depth, respectively.

As next step, the impact of the confounding factors SWC or nitrate concentration on the geophysical measurements was analyzed. Therefore, the $ECa^{EMI}$ of all six coil configurations, $EC^{ERT}$, and as reference the lab derived $ECe^{Soil}$ were correlated with the nitrate concentrations sampled and the SWC gathered by the sensors per treatment. The correlation with parameters per plot (soil vs geophysics) are additionally done for all treatment together. The color-coding and definition of quality of the
correlation coefficients $r$ are based on those suggested by Schmäck et al. (2022), whereby Schmäck et al (2022) used $R^2$ values instead of Pearson $r$. In general, for both EMI and ERT data the combined correlation where all data from all treatment were used together, with nitrate content and $ECe^{Soil}$ had a high to moderate correlation ($r > 0.4$) for the N fertilized plots. The low correlation for the control plot might be explained by the general low variability in the data (see Figure 9), but also clearly shows that all observed affects are mainly related to the fertilization and not to changes in other soil states. The low correlation
for the shallow soil layer (0 - 10 cm) over all EMI configuration, and also partly for the ERT, is somehow surprising, as especially in this layer most variation in nitrate concentration over time was observed (Figure 9). As the $ECe^{Soil}$ versus nitrate showed highest correlation, not only over the entire profile but also for the shallow layer 0-10 cm, except for the control, it shows that the geophysical sensors used (ERT and EMI) are less suited to gather information from the very shallow soil up to 10 cm. For ERT an electrode spacing of 0.25 cm was used, resulting in 1 m coverage of the 4-electrode used for 1 single
measurement (so partly influenced outside of the plots). Reducing the electrode spacing is theoretically feasible as shown by Ochs et al. (2020) but would lead to lower depth one can investigate and lower resolution at larger depths. Additionally, spatial smoothing is applied in the standard inversion used limiting the small-scale resolution as shown by Kemna et al. (2002). On the other hand, EMI derived ECa values are a weighted depth function of the retrieved signal, so one measurement spans laterally at least over the coil separation and the depth resolution. The sensitivity of the EMI configuration spans over the soil
sample interval and is therefore affected by different depths. Therefore, we can conclude that the shallow $ECa^{EMI}$ seem to be additionally affected by other soil states, and therefore, not suitable to measure the effect of fertilization in those shallow soil layer.



As generally known, and shown in Figure 10, the soil water content has a large impact on the soil electrical conductivity, $ECe^{Ref}$, measured by the sensors installed in the soil profile. Therefore, we also analyzed the impact on SWC on measured
$ECa^{EMI}$ and $EC^{ERT}$ by correlating the sensor based SWCs measured at the days of geophysical measurements against $ECa^{EMI}$ and $EC^{ERT}$ assuming that the SWCs are the same in all plots as those measured at the reference location. As can be seen in Table 2, the correlation is much better if we look over all depth and EMI configuration used, compared to the correlation of the $ECa^{EMI}$ with nitrate. Additionally, also the shallow soil depth 0-10 cm showed better correlation now. Only for the correlation between $EC^{ERT}$ and SWC no clear improvement can be detected, and one can even find a slightly worse correlation
between the two states. On the other hand, high correlations (as indicated in dark green in Table 2) do not show up for the correlation with SWC at all. Additionally, one can detect that the correlation is in most cases weaker for the high fertilizer application (Nc) compared to lower dosages applied, which corresponds to higher correlation found for the high nitrate concentrations (Nc) and electrical conductivity as discussed before. This means, that the soil water content has an overall impact on the measured EC at all fertilization levels but at high fertilization rates the nitrate impacts the measured EC to a
larger extend. Finally, the question arises why the shallow soil layer (0 - 10 cm) were better correlated to SWC compared to nitrate concentration. On one hand, the SWCs were all measured at one location (sensor pit) ignoring local soil heterogeneities, whereas the correlation with nitrate concentrations is based on two measurements within the same plot. Potentially the high variability (as expressed in high coefficient of variation) of the nitrate concentration might affected the correlation between nitrate and the geophysical measurements. Additionally, looking at the temporal course of $EC^{EMI}$ data (Figure 8), it was
observed that at days with low SWC the measured electrical conductivity dropped (e.g., around DAF 66 or 310), whereas the corresponding nitrate concentration dropped to a smaller extent as the nitrate mass per soil volume is not directly connected to the mass of water in the same soil volume. Nevertheless, the pore water conductivity must increase if nitrate masses stay unaffected, but SWC drops. This leads to the conclusion that at low SWC content, the EMI and ERT method seems not to be appropriate to detect changes in nitrate concentrations.

Based on the correlation between the obtained electrical conductivities and depth dependent nitrate concentrations, especially over all treatments, one can conclude, that high nitrate levels can be measured with moderate to higher accuracy in the clay soil studied. This also means that large differences in the nitrate level between fields or within fields might lead to differences in gathered $ECa^{EMI}$. This fact can either support the delineation of inner-field management zones or complicate explaining 'jumps' in measured $ECa^{EMI}$ across field boarders even if the same crop has been grown. That nitrate differences in the soil
can be measured by EMI has been already shown by Eigenberg et al. (2002) and Eigenberg and Nienaber (2003). In contrast to the work presented here, both studies did not only added nitrate to the soil but analyzed situations that are more complex. In Eigenberg et al. (2002) a seven-year manure and cover crop experiment was analyzed, where both, the manure but also the cover crops impact also soil organic matter content, soil structure, and water retention, and therefore, measured $ECa^{EMI}$. In the second study, Eigenberg and Nienaber (2003) studied the impact of compost piles on measured $ECa^{EMI}$ but also here various
changes in the soil (higher SOC, retention capacity etc.) might have affected the signal and not only nitrate stock differences.



## 4 Summary and Conclusion

In the study presented, time-lapse geophysical measurements using EMI, ERT, and GPR were used to analyze the impact of fertilization on the measured geophysical signals. Therefore, two different fertilizers (calcium ammonium nitrate and potassium chloride (KCL)) were applied at three different dosages on field plots in triplicates. Although, the GPR data could

not be used for a detailed analysis because of the high attenuations of the EM waves, for other soil types, where the background electrical bulk conductivity is smaller, GPR could help to map the SWC distributions in the soil over time. The results of this work showed that ERT was suitable to detect the impact of fertilization on bulk electrical conductivity over the entire course of the experiment (438 days) especially for the case of extremely high fertilizer application (10-fold of the recommended dosage). EMI was also not able to measure differences between the low fertilizer dosage applied (190 kg N per ha) and the

unfertilized control but for higher dosages EMI was able to trace the differences in electrical conductivity over longer time periods. Interestingly, EMI was able to detect KCL with 190 kg/ha. Correlation between EMI and ERT derived electrical conductivities and ground truth nitrate data showed that both techniques failed to trace the fertilizers back in the shallowest soil layer 0 - 10 cm and also were not able to reproduce small changes in nitrate concentrations in the unfertilized control plot. Based on the study, we can conclude, that EMI and ERT are suitable to detect different fertilization levels in a clay soil if the

fertilization differences are large enough but fail if concentrations are extremely small as those in a 10-year bare soil without fertilization. Nevertheless, the study showed that the geophysical measurements are suitable to detect differences in fertilization even over a longer time span of more than 1 year. In addition, we observed that relatively high SWC conditions are the most appropriate conditions to identify differences in fertilization states. Based on the findings presented, we recommend not neglecting the past fertilization practices in EMI studies, especially if larger areas are surveyed with different

fertilization practices or crops with different fertilizer demands.

*Competing interests.* The contact author has declared that none of the authors has any competing interests.

*Acknowledgement.* We acknowledge BMBF "BONARES", project Soil³ (grant 031B0026C). Furthermore, we thank the Terrestrial Environmental Observations (TERENO) for the support on the test site and of meteorological data. We would like to thank Philip. Steinberger, Durra Handri, Lena Lärm, Sabine Hasse, Jessica Schmäck, and Michael Iwanowitsch for their

help in the field.





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



**FIGURES AND TABLES**

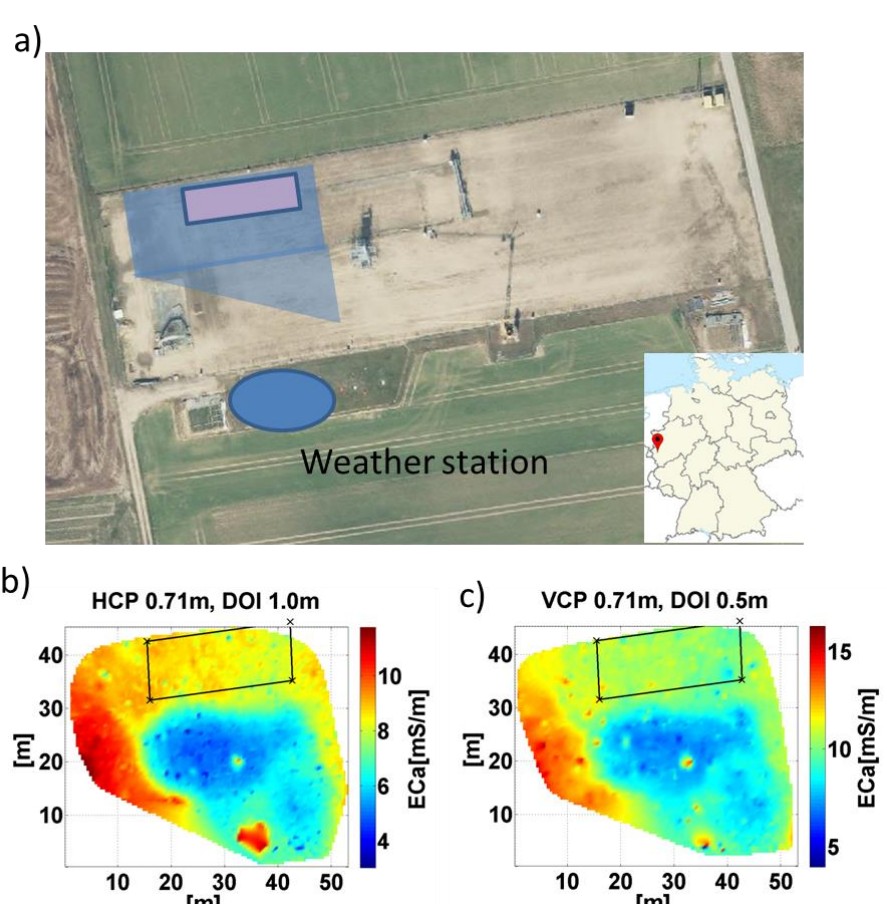

**Figure 1: a) Overview of the Selhausen test site (map adapted from © Google Earth, 2017). The location of the weather station and the soil sensors are marked with a blue and red circle, respectively, while the pre-experimental EMI survey area is shown by the blue plane. EMI measured ECa maps [mS/m] for b) HCP 0.71 m and c) VCP 0.71 m mode. The pink and black rectangular indicate the domain in which the field trails with the 21 plots as setup.**



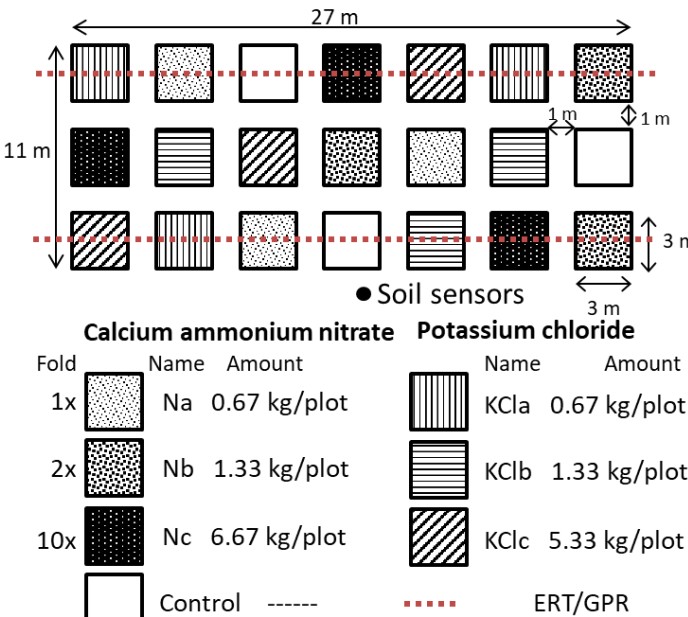

**Figure 2: Sketch of the experimental setup with control treatment (and the different dosages of calcium ammonium nitrate (N) and potassium chloride (KCl). Small letters a, b, and c indicate dosages. ERT and GPR lines are indicated with the red dashed lines and the location of the in-situ soil sensors is indicated with a black dot. Upper transect is transect 1 and lower transect 2.**



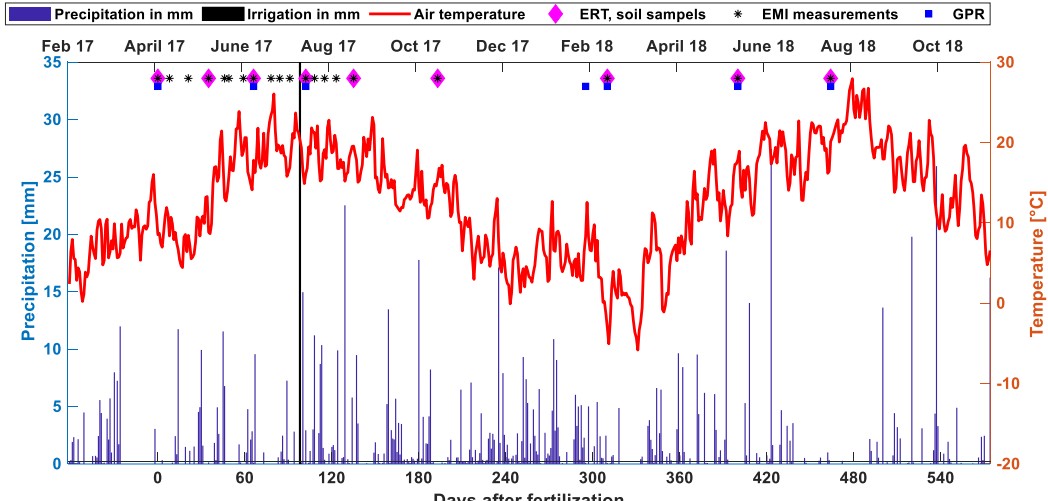

**Figure 3: Daily precipitation and daily mean air temperature measured by TERENO climate station SE_BDK_002. The EMI, GPR and ERT measurement and ground truth sampling are indicated by pink squares, black stars and blue boxes.**

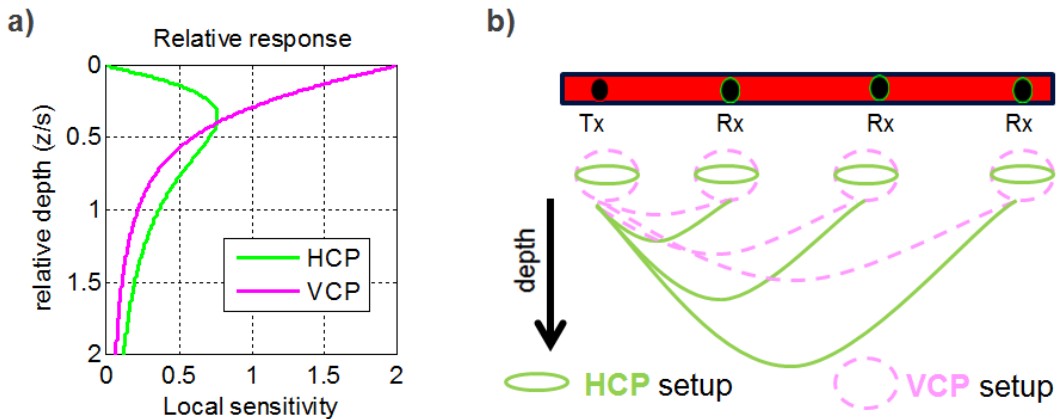

**Figure 4: a) Sensitivity response function after McNeill (1980) for the VCP and HCP EMI configuration with relative depth normalized by the coil separation. b) Sketch of a multi-coil MiniExplorer instrument, housing one transmitter and three receiver coils.**



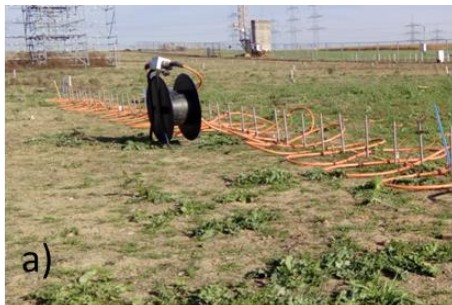

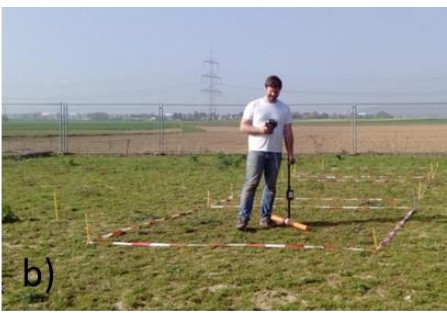

**Figure 5: a) ERT setup of one transects across 7 plots and b) EMI point measurement at on of the treatment plots.**

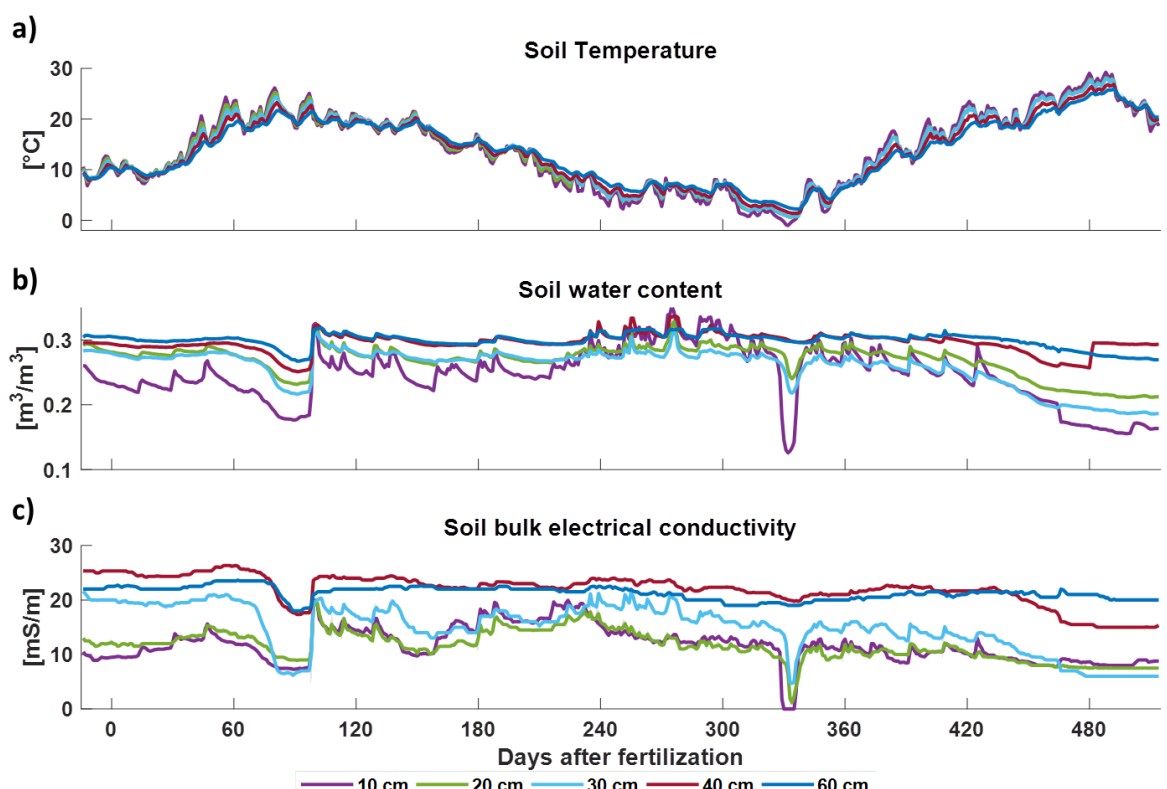

**Figure 6: Daily mean a) soil temperature [°C], b) volumetric soil water content [m³/m³], and c) soil temperature corrected (25 °C) soil bulk electrical conductivity ECeref [mS/m] for the depths 10, 20, 30, 40, and 60 cm.**





**Figure 7: a-f).** Selected inverted ERT profiles (not temperature corrected) showing the effect of fertilization over time on the $EC^{ERT}$ values. The location of the different plots is indicated at the top of the figure (see Figure 2 for more information). g-i) GPR common offset profiling (COP) data at selected dates. The top pictures in the right row show the radargrams, where high and low amplitudes are indicated by white and black color. Below are depicted the corresponding hilbert envelope images of the same measurement day.



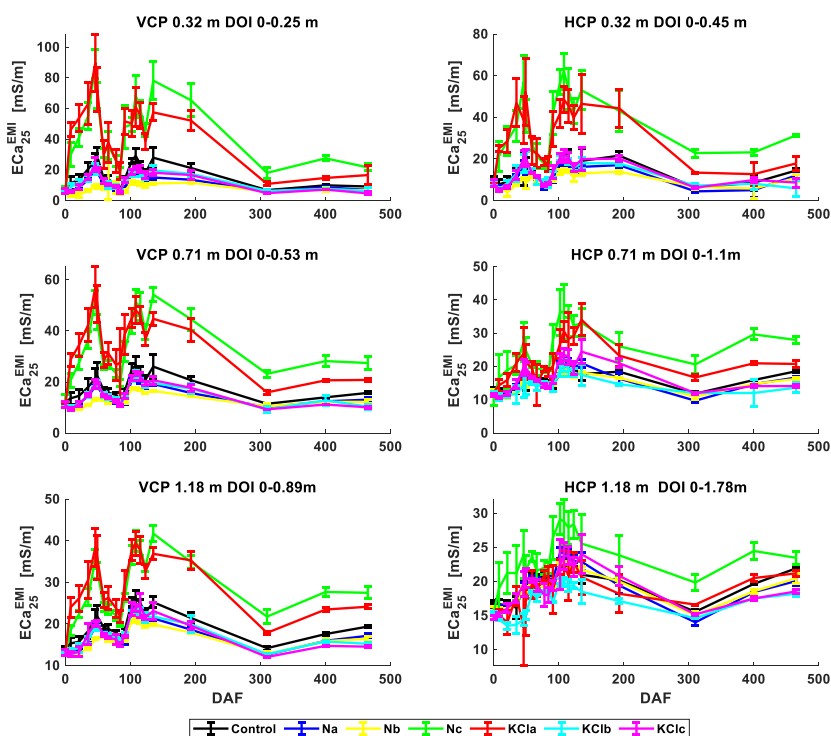

**Figure 8: Changes ECa^EMI over times for the six different EMI configurations, value represent the mean per treatment and the crossbar indicates the standard deviation. Colors represent the different fertilizer treatments. Note for better visualization different limits in the y-axis are used.**



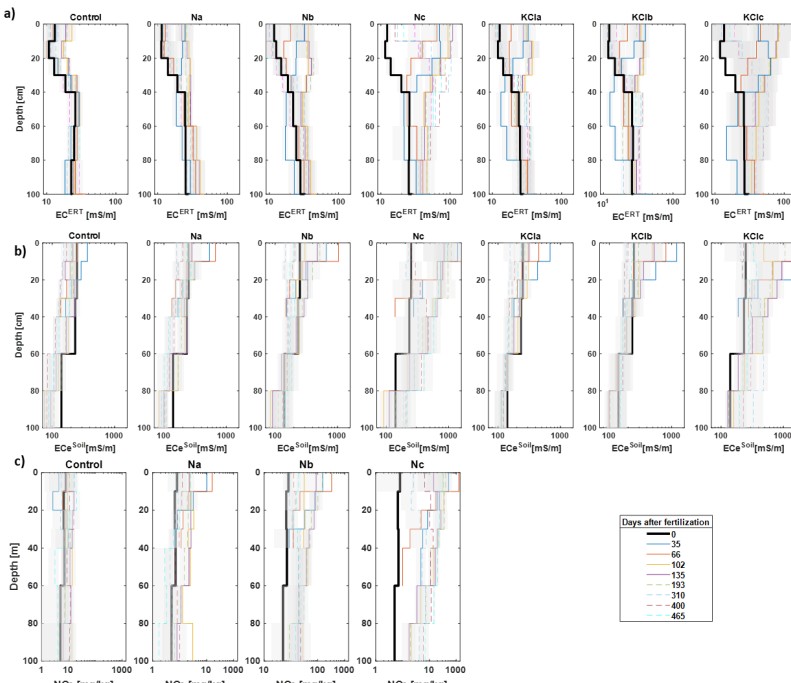

**Figure 9: Comparison of the mean a) EC$^{ERT}$ [mS/m], b) EC$_e^{Soil}$ [mS/m], and c) nitrate concentration [mg/kg] for each treatment over time. As reference the depth profile prior to fertilization is shown in black. The grey color indicates the standard deviation. Note that the x-axes scales is logarithmic for better visualization.**



TABLES

**Table 1: Overview of all measured and methods.**

|  | Measured parameters | Temporal resolution | Spatial resolution |
|---|---|---|---|
| climate data | Air temperature [°C]<br><br>Precipitation [mm] | Continuous measurements<br>Daily average | Reference location see Figure 1 |
| Soil sensors | Soil water content [m³/m³]<br><br>Soil temperature [°C]<br><br>Soil bulk electrical conductivity (ECe$^{ref}$) [mS/m] | Continuous measurements<br>Daily average | Reference location see Figure 1<br>Sensors depth 10, 20, 30, 40, and 60 cm |
| Soil sampling | Nitrat content (NO$_3^-$) [mg/kg]<br><br>Soil bulk electrical conductivity (ECe$^{Soil}$) [mS/m] | Total 9 dates | 1 auger per plot,<br>7 depths<br>(0-10, 10-20,20- 30, 30-40,40-60,60- 80,80-100 cm) |
| EMI | Apparent electrical conductivity (ECa$^{EMI}$) [mS/m]<br><br>with 6 configurations:<br><br>VCP 0.32, VCP0.71, VCP1.18, HCP0.32, HCP0.71, HCP1.18,<br><br>(Orientation + coil separation in m) | Frequently until DAF 130, afterwards on same dates as soil samples<br><br>Total 20 | 4 measurements per plot |
| ERT | Inverted electrical conductivity (EC$^{ERT}$) [mS/m] | Total 9 | 2 profiles by each 30 m<br>(2 reputation per treatment)<br>7 Depths for correlation:<br>0-10, 10-20, 20-30, 30-40, 40-60, 60-80, 80-100 cm |
| GPR | Relative comparison | Total 7 | 3 transects every 30 cm |





**Tabel 2: Pearson correlation coefficients ($r$) between ECa$^{EMI}$ (VCP and HPC), EC$^{ERT}$, and ECe$^{Soil}$ with the measured a) nitrate concentrations and b) SWC data from the sensors for depths (d) between 10 to 120 cm. Note, that the mean values gather over all plots within the same treatments for each measurement day have been used. Correlation was color coded according to $r$, with high correlation coefficients in black bold underlined ($r > 0.8$), moderate correlation in black bold ($0.5 < r > 0.8$), low correlation in black ($0.3 < r > 0.5$) and no correlation in grey ($r < 0.3$). The definition of quality of the correlation coefficients $r$ are similar than those defined by Schmäck et al. (2022).**

a) Nitrate          b) SWC

| | $d$ [m] | 10 | 20 | 30 | 40 | 60 | 80 | 100 | | 10 | 20 | 30 | 40 | 60 |
|---|---|---|---|---|---|---|---|---|---|---|---|---|---|---|
| ECa$^{EMI}$ VCP 0.32 DOI 0-0.25 m | All | 0.53 | **0.83** | 0.76 | 0.67 | 0.55 | 0.38 | 0.20 | | | | | | |
| | Co. | -0.06 | -0.16 | -0.09 | -0.02 | 0.50 | 0.18 | 0.14 | | 0.68 | 0.58 | 0.59 | 0.63 | 0.58 |
| | Na | -0.04 | 0.26 | 0.54 | 0.47 | 0.73 | 0.70 | 0.35 | | 0.69 | 0.67 | 0.63 | 0.56 | 0.53 |
| | Nb | 0.30 | 0.71 | 0.76 | 0.58 | 0.60 | 0.13 | -0.08 | | 0.63 | 0.57 | 0.58 | 0.50 | 0.45 |
| | Nc | 0.21 | 0.62 | 0.51 | 0.30 | 0.15 | -0.17 | -0.23 | | 0.51 | 0.45 | 0.45 | 0.34 | 0.27 |
| ECa$^{EMI}$ VCP 0.53 DOI 0-0.53 m | All | 0.48 | **0.86** | **0.81** | 0.75 | 0.64 | 0.46 | 0.28 | | | | | | |
| | Co. | -0.05 | -0.03 | 0.08 | 0.03 | 0.54 | 0.35 | 0.16 | | 0.59 | 0.56 | 0.61 | 0.62 | 0.60 |
| | Na | 0.02 | 0.35 | 0.71 | 0.63 | 0.78 | 0.72 | 0.40 | | 0.67 | 0.66 | 0.67 | 0.64 | 0.62 |
| | Nb | 0.29 | 0.70 | 0.78 | 0.62 | 0.64 | 0.14 | 0.01 | | 0.59 | 0.58 | 0.61 | 0.55 | 0.51 |
| | Nc | 0.14 | 0.72 | 0.65 | 0.47 | 0.31 | -0.09 | -0.22 | | 0.57 | 0.48 | 0.49 | 0.46 | 0.40 |
| ECa$^{EMI}$ VCP 1.18 DOI 0-0.89 m | All | 0.43 | **0.83** | **0.81** | 0.78 | 0.68 | 0.51 | 0.35 | | | | | | |
| | Co. | -0.05 | -0.02 | 0.10 | 0.09 | 0.56 | 0.40 | 0.20 | | 0.66 | 0.59 | 0.63 | 0.67 | 0.63 |
| | Na | -0.08 | 0.30 | 0.74 | 0.68 | 0.78 | 0.74 | 0.38 | | 0.75 | 0.68 | 0.70 | 0.71 | 0.67 |
| | Nb | 0.23 | 0.69 | 0.80 | 0.66 | 0.67 | 0.18 | 0.08 | | 0.67 | 0.61 | 0.64 | 0.63 | 0.58 |
| | Nc | 0.04 | 0.68 | 0.67 | 0.57 | 0.40 | 0.00 | -0.10 | | 0.66 | 0.50 | 0.50 | 0.55 | 0.47 |
| ECa$^{EMI}$ HCP 0.32 DOI 0-0.45 m | ALL | 0.43 | **0.87** | **0.83** | 0.77 | 0.65 | 0.45 | 0.46 | | | | | | |
| | Co. | -0.03 | 0.04 | 0.12 | 0.14 | 0.38 | 0.23 | -0.05 | | 0.40 | 0.31 | 0.37 | 0.39 | 0.36 |
| | Na | -0.05 | 0.28 | 0.68 | 0.64 | 0.66 | 0.67 | 0.30 | | 0.56 | 0.52 | 0.54 | 0.51 | 0.49 |
| | Nb | 0.18 | 0.69 | 0.79 | 0.70 | 0.67 | 0.15 | 0.00 | | 0.58 | 0.53 | 0.56 | 0.52 | 0.48 |
| | Nc | 0.07 | 0.78 | 0.69 | 0.55 | 0.36 | -0.09 | -0.26 | | 0.57 | 0.47 | 0.47 | 0.47 | 0.41 |
| ECa$^{EMI}$ HCP 0.53 DOI 0-1.1 m | All | 0.30 | **0.80** | **0.81** | **0.84** | 0.75 | 0.63 | 0.00 | | | | | | |
| | Co. | 0.12 | 0.32 | 0.16 | 0.01 | 0.32 | 0.32 | 0.00 | | 0.50 | 0.36 | 0.40 | 0.50 | 0.48 |
| | Na | -0.07 | 0.20 | 0.84 | 0.75 | 0.69 | 0.62 | 0.53 | | 0.63 | 0.52 | 0.55 | 0.62 | 0.60 |
| | Nb | 0.09 | 0.63 | 0.79 | 0.79 | 0.71 | 0.21 | 0.09 | | 0.65 | 0.53 | 0.56 | 0.61 | 0.55 |
| | Nc | -0.14 | 0.65 | 0.71 | 0.77 | 0.61 | 0.29 | 0.13 | | 0.61 | 0.36 | 0.33 | 0.53 | 0.46 |
| ECa$^{EMI}$ HCP 1.18 DOI 0-1.78 m | All | 0.30 | 0.67 | 0.70 | 0.72 | 0.62 | 0.57 | 0.47 | | | | | | |
| | Co. | -0.15 | -0.10 | 0.01 | -0.08 | 0.39 | 0.36 | 0.27 | | 0.65 | 0.56 | 0.59 | 0.65 | 0.58 |
| | Na | -0.08 | 0.10 | 0.76 | 0.67 | 0.64 | 0.63 | 0.51 | | 0.69 | 0.59 | 0.63 | 0.70 | 0.64 |
| | Nb | 0.04 | 0.53 | 0.63 | 0.70 | 0.70 | 0.35 | 0.28 | | 0.71 | 0.59 | 0.61 | 0.70 | 0.62 |
| | Nc | -0.09 | 0.45 | 0.55 | 0.56 | 0.41 | 0.21 | 0.15 | | 0.59 | 0.37 | 0.33 | 0.53 | 0.45 |
| ECa$^{ERT}$ | All | 0.60 | **0.89** | 0.71 | 0.74 | 0.70 | 0.62 | 0.45 | | | | | | |
| | Co. | -0.18 | 0.02 | 0.06 | 0.13 | 0.35 | 0.09 | -0.05 | | 0.63 | 0.41 | 0.32 | 0.20 | 0.16 |
| | Na | -0.10 | 0.39 | 0.87 | 0.78 | 0.68 | 0.23 | 0.09 | | 0.39 | 0.36 | 0.36 | 0.15 | 0.09 |
| | Nb | 0.34 | 0.63 | 0.82 | 0.89 | 0.66 | 0.11 | 0.54 | | 0.33 | 0.45 | 0.50 | 0.03 | 0.03 |
| | Nc | 0.39 | 0.79 | 0.40 | 0.47 | 0.47 | 0.46 | 0.34 | | 0.28 | 0.31 | -0.04 | 0.86 | 0.81 |
| ECe$^{Soil}$ | All | 0.97 | 0.94 | 0.98 | 0.96 | 0.90 | 0.94 | 0.58 | | | | | | |
| | Co. | -0.23 | -0.19 | 0.01 | 0.11 | 0.18 | -0.41 | -0.62 | | | | | | |
| | Na | 0.90 | 0.53 | 0.38 | 0.56 | 0.71 | 0.59 | -0.65 | | | | | | |
| | Nb | 0.97 | 0.86 | 0.82 | 0.77 | 0.59 | 0.57 | 0.02 | | | | | | |
| | Nc | 0.97 | 0.92 | 0.98 | 0.97 | 0.85 | 0.96 | 0.87 | | | | | | |