# Peer review of "ASSESSING SOIL FERTILIZATION EFFECTS USING TIME-LAPSE ELECTROMAGNETIC INDUCTION"

_EGUsphere, 2024_

## Referee Comment (RC1)

Assessing soil fertilization effects with near surface geophysics is a very interesting topic and the authors' performed a nice controlled experiment to comprehend the suitabilities of the sensing technologies, especially electromagnetic induction. I am confident this work would benefit the readers of the proximal soil sensing and agrogeophysics communities. I recommend a moderate revision of the manuscript as a figure is missing and I have a few minor suggestions to improve the readability.

Scientific comments:

1) Have you considered inverting the ECa data from EMI? Why was this not done when it would improve the overall analysis? Please elaborate. I think it would be nice to show in Fig. 7 along with the ERT and GPR profiles.
2) Fig. 10 is missing. So, I did not manage to follow section 3.2.3 completely.

General comments:

3) I request the authors' to re-read the manuscript a few times to improve the grammar and correct the spelling mistakes.
4) Some of the paragraphs are very big and need to be split into multiple paragraphs to improve the readability. For example, the first paragraph is too big. Please split it in to two. A suggestion is to split where you start discussing about EMI.
5) Please abbreviate recurring terms such as apparent electrical conductivity as ECa and use the same terminology consistently.

Specific comments:

1) In lines 20-30, please list the important findings with numbers 1, 2, 3. It is slightly confusing with the usage of "On the other hand.." twice.
2) In line 97 and 352, "extend" should be "extent".
3) In Fig. 2, it would be nice if you include the profile picture of soil sensor installation.
4) In line 130-135, please rephrase "Over a time period of 485 days…." sentence. It reads as if the EMI data was collected everyday.
5) In line 175, change "followed data inversion" to "following data inversion".
6) In line 337, "SCW" should be "SWC".
7) In Table 2, replace $ECa^{ERT}$ with $EC^{ERT}$.
8) In Fig. 6, please consider also showing the dates on the top x axis.

All the best!

---

## Author Comment (AC1)

Dear Editors and Reviewers,

Thank you very much for your detailed and helping review of the manuscript.

We have processed all the comments and below you can find a detailed list of our response in blue on the raised comments.

Yours sincerely,

Anja Klotzsche & Manuela Kaufmann on behalf of the author team

RC1:

Assessing soil fertilization effects with near surface geophysics is a very interesting topic and the authors' performed a nice, controlled experiment to comprehend the suitabilities of the sensing technologies, especially electromagnetic induction. I am confident this work would benefit the readers of the proximal soil sensing and agrogeophysics communities. I recommend a **moderate revision** of the manuscript as a figure is missing and I have a few minor suggestions to improve the readability.

Scientific comments:

1) Have you considered inverting the ECa data from EMI? Why was this not done when it would improve the overall analysis? Please elaborate. I think it would be nice to show in Fig. 7 along with the ERT and GPR profiles.
Thanks for the comment. Yes we considered inverting the EMI data and we also did and tested the performance. In a first step we tested different inversion options with different layer thicknesses and number of layers (between 2-4 layers). Unfortunately, the results from the inversion where the layer thickness was estimated were difficult to be used in a regression analysis due to different layer numbers/thicknesses. Therefore, in a second step, we kept the layer thicknesses constant and only inverted for the conductivity. The soil layers were the same as for the soil sensors with 8 layers. The results were not satisfying since only 6 measurements were not enough to constrain the number of layers. For our experiment, we didn't think that the inversion provides any added value, especially as EMI data are often directly used and not being inverted.

2) Fig. 10 is missing. So, I did not manage to follow section 3.2.3 completely.
We apologize for the inconvenience, and we now added the missing figure.

General comments:

3) I request the authors' to re-read the manuscript a few times to improve the grammar and correct the spelling mistakes.
We apologize for the small mistakes and checked the text carefully again. We hope that spelling and grammar is not correct.

4) Some of the paragraphs are very big and need to be split into multiple paragraphs to improve the readability. For example, the first paragraph is too big. Please split it in to two. A suggestion is to split where you start discussing about EMI.
We went through the manuscript and considered this comment and split the longer paragraphs where it seemed to be logic.

5) Please abbreviate recurring terms such as apparent electrical conductivity as ECa and use the same terminology consistently.

We agreed and used abbreviations wherever possible and corrected misleading abbreviations.

Specific comments:

1) In lines 20-30, please list the important findings with numbers 1, 2, 3. It is slightly confusing with the usage of "On the other hand." twice.

We changed the text to: "The results showed that 1) the commonly used CAN application dosage did not impact the geophysical signals significantly. 2) EMI and ERT were able to trace back the temporal changes in nitrate concentrations in the soil profile over more than one year. 3) Both techniques were not able to trace the nitrate concentrations in the very shallow soil layer of 0 – 10 cm. Irrespectively of the low impact of fertilization on the geophysical signal. 4) The results indicated that past fertilization practices cannot be neglected in EMI studies, especially if surveys are performed over large areas with different fertilization practices or crop grown with different fertilizer demands or uptake." Line 23-29

2) In line 97 and 352, "extend" should be "extent".
Done as suggested.

3) In Fig. 2, it would be nice if you include the profile picture of soil sensor installation.
We added an image of the soil pit and adapted the figure description.

4) In line 130-135, please rephrase "Over a time period of 485 days…." sentence. It reads as if the EMI data was collected everyday.
We rephrased it to "To monitor the effect of fertilization over time, 20 EMI measurements were conducted over a period of 485 days (DAF 485) (see Table 1)." See line 134-135.

5) In line 175, change "followed data inversion" to "following data inversion".
Done as suggested.

6) In line 337, "SCW" should be "SWC".
Done as suggested.

7) In Table 2, replace ECa ERT with EC ERT.
Done as suggested.

8) In Fig. 6, please consider also showing the dates on the top x axis.
We updated Figure 6 and added the x-axis on top.

All the best!

---

## Author Comment (AC2)

Dear Editors and Reviewers,

Thank you very much for your detailed and helping review of the manuscript.

We have processed all the comments and below you can find a detailed list of our response in blue on the raised comments.

Yours sincerely,

Anja Klotzsche & Manuela Kaufmann on behalf of the author team

RC2:

The study of fertilisers in precision agriculture is crucial. We know that geophysical techniques can be of great help to agricultural practices and this work demonstrates once again their efficient use. I believe that this article is of interest to the agrogeophysical community. I recommend a **minor revision**, there are a few things to revise in the text and especially the lack of a figure did not allow a careful reading of the final part of the paper.

Specific comments:

1. Line 112: "The main textural fraction is silt with 55-67% silt in all horizons". The word silt is repeated too many times in such a short sentence. Suggestion: "The main textural fraction, accounting for 55–67% across all horizons, is silt".
   Thanks, we changed it according to your suggestion.

2. Figure 1: From the text (lines 115-119) and the caption, it is not clear in Figure 1 the black box, and the red circle. I suggest either revising the caption or aligning the text with what is in Figure1.
   Thanks for the hint. We changed the caption to "… In the lower left of a) is the location of the test site in Germany marked with a red dot. EMI measured ECa maps [mS/m] for b) HCP 0.71 m and c) VCP 0.71 m mode. The pink box in a) and black rectangular in b) & c) indicate…"

3. Figure 2: It might be clearer to write the letters a, b and c as subscripts. I also suggest inserting a small number to indicate plot 1 and 21
   Dones as suggested. Similary we adapted the text according to this.

4. Line 131: Rephrase the sentence. EMI data were run after 485 days and not every day.
   We rephrased the sentence to "To monitor the effect of fertilization over time, 20 EMI measurements were performed over a period of 485 days (DAF 485) (see Table 1)." See line 134-135.

5. Line 190: Perhaps reference is made to figure 2 and not figure 1.
   Yes indeed. Thanks. We corrected it.

6. Line 250-270: For clarity, add the reference to Figure 7 in the text for images b-f and h-i as well.
   Thanks for this comment. We adapted the text according to it.

7. Line 337: Replace SCW with SWC.

Done as suggested

General comments:

1. I suggest the authors reread the text, there are some grammatical and spelling errors.
   We apologize for the inconvenience and checked the text again We have grammar and spelling is not correct.

2. There are many terms in the text that are sometimes abbreviated. It would be appropriate to use abbreviations from the outset and to use them throughout the text without repeating long terminology. For example, use the abbreviation ECa for Electrical Conductivity from the introduction. The abbreviation VCP and HCP appears first in Figure 1 cited in section 2.1 and then the extended terminology is given later in section 2.4.
   Same has been requested by reviewer #1 and we tried to use abbreviations throughout the text now wherever feasible.

3. Tables and figures are a bit small.
   We apologize and hope that in the final printed version all figures and tables are set in a readable size.

4. Figure 10 is missing; therefore it was difficult to follow section 3.2.3.
   We apologize for the inconvenience and we now added the missing figure.

5. Considering that the EMI technique measures the electrical conductivity of the soil, a comparison between this parameter and the conductivity derived from ERT measurements could be useful.
   We compared the values and as except an offset was observed. Since we do not calibrate and invert the EMI data, we should not directly compare the different ECa values. The sensitive of the EMI and ERT is also depending on the configuration quite different und a certain difference is expected.